# Working Memory in Children with Learning Disorders: An EEG Power Spectrum Analysis

**DOI:** 10.3390/brainsci10110817

**Published:** 2020-11-04

**Authors:** Benito J. Martínez-Briones, Thalía Fernández-Harmony, Nicolás Garófalo Gómez, Rolando J. Biscay-Lirio, Jorge Bosch-Bayard

**Affiliations:** 1Departamento de Neurobiología Conductual y Cognitiva, Instituto de Neurobiología, Universidad Nacional Autónoma de México Campus Juriquilla, Querétaro 76230, Mexico; benjavik332@gmail.com (B.J.M.-B.); thaliafh@yahoo.com.mx (T.F.-H.); 2Instituto de Neurología y Neurocirugía, La Habana 10400, Cuba; nicogaro72@gmail.com; 3Centro de Investigación en Matemáticas, Guanajuato 36000, Mexico; 4McGill Centre for Integrative Neuroscience (MCIN), Ludmer Centre for Neuroinformatics and Mental Health, Montreal Neurological Institute (MNI), McGill University, Montreal, QC H3A2B4, Canada

**Keywords:** learning disorders, working memory, school-age children, EEG power spectral density, source localization, sLORETA

## Abstract

Learning disorders (LDs) are diagnosed in children whose academic skills of reading, writing or mathematics are impaired and lagging according to their age, schooling and intelligence. Children with LDs experience substantial working memory (WM) deficits, even more pronounced if more than one of the academic skills is affected. We compared the task-related electroencephalogram (EEG) power spectral density of children with LDs (*n* = 23) with a control group of children with good academic achievement (*n* = 22), during the performance of a WM task. sLoreta was used to estimate the current distribution at the sources, and 18 brain regions of interest (ROIs) were chosen with an extended version of the eigenvector centrality mapping technique. In this way, we lessened some drawbacks of the traditional EEG at the sensor space by an analysis at the brain-sources level over data-driven selected ROIs. Results: The LD group showed fewer correct responses in the WM task, an overall slower EEG with more delta and theta activity, and less high-frequency gamma activity in posterior areas. We explain these EEG patterns in LD children as indices of an inefficient neural resource management related with a delay in neural maturation.

## 1. Introduction

Learning disorders (LDs) are a main neurodevelopmental impairment, with a prevalence of 5–15% in children between 5 and 16 years old [1,2,3]. A specific LD is diagnosed for persistent difficulties in learning academic skills such as reading, writing or mathematics. The LD diagnosis also requires the impaired academic skill to be significantly lagging for the age, schooling and the Intelligence Quotient (IQ) of the child [2]. Moreover, an LD child with a combined deficiency in two or three of these skills is a frequently found subtype of LD, formerly known as LD not otherwise specified [4], and this co-occurrence of academic impairments appears in between 30 and 70% of the LD cases in children [5].

Learning disorders usually include a heterogeneous frame of cognitive impairments [5]. A known source of this heterogeneity is a working memory (WM) deficit [6]. Working memory is the capacity to store and manipulate information for short periods of time at the service of goal-directed behaviors [7] and is the consistently most affected cognitive domain in LD children [8,9,10]. Indeed, WM performance not only distinguishes properly between LD and children with typical development [11]; it is also an adequate predictor of future academic difficulties [12], and is more severely affected in LD children with more than one academic skill impaired (LD not otherwise specified) [5].

Neural substrates of LD have been identified with quantitative electroencephalogram (qEEG) analyses [13]. The qEEGs of LD children show an abnormally slower resting-state activity compared to age-matched controls. This slower activity is akin to that of younger healthy subjects since it involves more theta activity in frontal regions and less alpha power in posterior (parietal and occipital) regions, like in previous developmental stages of the healthy child. For this reason, LDs are considered developmental disorders with a delay in EEG maturation that impairs the ability to keep up with normal peers at school [14,15,16].

The neural substrates of cognitive performance, particularly WM, have been examined with mainstream EEG techniques such as event-related potentials, power spectral density (PSD) analyses, and connectivity measures (e.g., coherence). In healthy adults, task-related PSD findings (while the individual performs a cognitive task) have revealed an increased delta activity that has been implied in states of sustained concentration coupled with the inhibition of sensory information [17,18,19,20]. For the theta band, a main finding is an increased theta activity at frontal sites coupled with a decreased global alpha power compared to a resting-state or a tonic condition [17,21,22,23], and the theta increase is even more pronounced for higher WM loads, with more items to memorize [24,25]. The higher task-related theta power has been broadly regarded as an attentional control mechanism not specifically linked to WM, being more pronounced in less apt individuals, at greater task difficulties, and in conditions in which the focusing of attention requires more effort; thus, the task-related theta power is considered to be increasingly affected according to the neural resources needed to properly perform a cognitive task [22,24,26,27,28,29]. As for the role of alpha activity in memory, this frequency band has remained more equivocal, with its initial recognition as a mere idling state waiting to be suppressed during cognitive effort [30]. However, both the theta and lower-alpha power were found to be increased during the performance of a WM task [31], and a greater alpha power, mainly of the upper 10–12 Hz range in posterior regions has been implied, with an inhibitory top-down control role of blocking the processing of irrelevant stimuli [32,33,34,35]. Conversely, a pronounced decrease in the power of the upper-alpha band, also known as an event-related desynchronization (ERD), has been established to reflect cortical activation instead of inhibition [36,37]. The upper-alpha ERD has also been related with the neural efficiency hypothesis, which states that a more efficient brain functioning involves less and more focused brain activation, since relationships have been observed in healthy adults between a higher IQ and an adequate task performance that involves less neural activity, including a lower brain hemodynamic response [38,39]. In WM tasks, a better performance has been related with a higher upper-alpha power or, conversely, with less cortical activation, indicated by lower levels of ERD [40,41], but there have also been results that do not support this hypothesis, showing the opposite pattern of an upper-alpha ERD [42] and even a higher theta power in brighter individuals [43,44]. For the higher frequencies of beta and gamma; a higher beta power (14–28 Hz) has been associated with a subvocal rehearsal during the retention of items [45], and a higher upper-beta (20–30 Hz), with the preparation of motor responses [17]. In addition, a sustained increase in gamma activity over posterior sites is involved, with a role in memory maintenance and the binding of memory representations [46,47,48,49,50].

In the case of task-related EEG research in children, a study with dyslexic children who responded to a phonological discrimination task [51] found the children had a higher frontal theta power over healthy controls, a finding explained as compensatory for an inefficient attentional control due to cortical immaturity. Spironelli, Penolazzi and Angrilli [52] also found a slower and greater theta activity peak in the right hemisphere of dyslexic children, coupled with an insufficient left hemisphere theta activity. However, Klimesch (2001) [53] examined dyslexic children vs. controls in a reading task and found the dyslexics had less frontal theta activity and the control children had a more selective left hemisphere recruitment of theta power at posterior sites. As for children performing WM tasks, a study that compared the PSD of healthy children with adults [18] found the children had more delta, more theta, and less alpha activity, occurrences also interpreted as compensatory mechanisms due to neural immaturity. Moreover, work that compared poor readers vs. normal control children with an event-related potential analysis (ERP) in a Sternberg WM task [54] found the poor readers had longer and larger P300 latencies at frontal sites [55]; this points to a greater effort required by the LD children, since the P300 relative amplitude is considered a marker of the amount of attentional resources given to a cognitive task [56].

The previous work generally points to a greater recruitment of delta and theta power in more difficult task conditions, and in less mature (or more unfit) populations with greater difficulties in performing cognitive tasks. Conversely, according to the findings in healthy adults, the EEG indices of a normal WM functioning would imply the recruitment of the higher frequencies of beta and gamma.

To our knowledge, the WM (task-related) EEG power spectrum has not been explored in children with LD. This was the main goal of the present work. Specifically, we aimed to compare the EEG PSD of children with LD with a control group of children with good academic achievement during a Sternberg-type WM task. However, children with LD constitute a very heterogeneous population regarding their WM functioning [11]; therefore, we also included the IQ in the analysis as a multifactorial predictor to assess whether the Intelligence Coefficient (IQ) may influence the percentage of correct responses given by the children during the experiment.

We characterized the brain electrical activity (PSD) during the task in terms of the differences between the two groups in each condition and as the differences within each group separately for the two conditions. We performed this analysis using a multifactorial model, in which we included both the IQ and the percentage of correct responses (PCR) as multifactorial predictors [57]. In the case of the PCR, to guarantee the statistical requirement of balanced samples for this analysis, we divided the subjects in two groups according to their PCR as Good Responders (GR subjects), those with the highest percentage who fell within the top half of the PCR histogram, and Bad Responders (BR subjects), with those ones falling in the lower half of the PCR histogram. Previously mentioned findings (in healthy adults) about the neural efficiency hypothesis have identified a relationship between less upper-alpha ERD and a higher IQ, and a better cognitive performance of good responders [39,41]. Therefore, we also examined the relationship between the PSD, the type of responders and IQ considering this neural efficiency hypothesis.

In this work, we did not aim to include other possible neuropsychological variables that may have influenced the children’s response, since it was beyond the said focus of the present work.

The EEG analysis performed in most of the above-cited papers was performed on the EEG voltage at the sensor space (over the scalp), a classic approach with two main drawbacks: the volume conduction effect, and the reference electrode effect, both of which induce a mixing of signals in the electrodes that distorts the real neurophysiological events [58]. The volume conduction is a passive spread of electrical activity (independent of post-synaptic potentials) through the brain tissue, cerebrospinal fluid, skull and scalp. The reference electrode, on the other hand, affects things differently according to its location, causing a difference in electrical potential against the active electrodes of the scalp. Both shortcomings can be partially solved by spatial filtering measures (such as the surface Laplacian technique) and source localization methods. The latter methods require several assumptions about the current source localization that are even more critical with fewer electrodes employed [59]. However, source localization techniques such as eLoreta and sLoreta have several advantages that remarkably diminish the possible localization errors and overcome the sensor space limitations by analyzing the contribution of specific cortical brain areas [59,60]. On this basis, using sLoreta, we performed a power spectrum analysis of the estimated primary currents at the source level (instead of the voltage signals in the sensor space).

## 2. Materials and Methods

The Ethics Committee of the Instituto de Neurobiología, Universidad Nacional Autónoma de México (UNAM), approved the experimental protocols (INEU/SA/CB/145, INEU/SA/CB/146, 1 July 2015), which followed the Ethical Principles for Medical Research Involving Human Subjects established by the Declaration of Helsinki [61]. Informed consent was signed by all the children and their parents.

### 2.1. Participants

Forty-five children from 8 to 11 years of age were selected (see the inclusion criteria below), from a larger sample of over 100 children referred by social workers from several elementary schools in Querétaro, México. The sample was divided into two groups: 22 control children with good academic achievement (Ctrl group) and 23 children diagnosed with LD with problems in two or three academic domains (reading, writing and mathematics). Figure 1 shows the frequencies of the academic impairments.

All children fulfilled the following inclusion criteria: a normal neurological and psychiatric exam (except for the diagnostic requirements of the LD group), a low–average or greater Intelligence Quotient (IQ) [62], a parent (mother) with at least a completed elementary school education, and a per capita income greater than 50 percent of the minimum wage.

The LD diagnosis was established based on the following three criteria: (a) poor academic achievement reported by teachers and parents; (b) percentiles at 16 or lower on the subscales of reading, writing and mathematics of the Infant Neuropsychological Scale for Children [63]; and (c) LD diagnosis by a psychologist according to the DSM-5 criteria [2]. Several of them failed on different items in the attentional evaluation of the DSM-V, as is common in this disorder [64,65], but they did not meet the DSM-5 criteria for ADHD [2].

Children who belonged to the Ctrl group, in addition to showing good academic achievement, obtained percentiles of 26 or above in the reading, writing and mathematical domains of the Neuropsychological Scale for Children [63]. Table 1 shows the characteristics of both groups.

### 2.2. Working Memory Task

The WM task used in this work was a modified version of the classic Sternberg WM task [54]. A verbal WM task was employed since LD children impaired in two or three academic skills show a more consistent deficit in the phonological loop subsystem of Baddeley’s WM model [5,7].

The WM task (Figure 2) consisted of two conditions (low-load and high-load) presented in 180 trials, with 90 trials per condition appearing at random. In each trial, four digits appeared simultaneously on the screen. In the low-load condition, all the digits were the same; in the high-load condition, the digits were different. The participants were instructed to memorize the numbers, and after each trial, a single digit appeared; they had to press one button (match response) if the digit was included in that trial and press another button if not (non-match response). Between the four digits (stimuli to remember) and the single digit (stimulus to respond to), a segment of 800 ms was considered the main section of the retention phase of the task. In this segment, we performed power spectrum analysis for the trials with correct responses. The stimuli were presented with the software MindTracer [66] and synchronized with the EEG data acquisition system.

### 2.3. EEG Recording and Data Analysis

Children were comfortably seated in a dim-lit, faradized and soundproofed room. The EEG was recorded during the task performance using a Medicid IV system (Neuronic Mexicana, S.A.; Mexico City, Mexico) and Track Walker TM v5.0 data system, from 19 leads of the 10–20 system (ElectroCap, International Inc.; Eaton, OH, USA) referenced to the linked earlobes (A1–A2). The amplifier bandwidth was set between 0.1 and 50 Hz. The signal was amplified with a gain of 20,000, with electrode impedances at or below 5 kΩ. The EEG data were sampled every 5 ms with a sampling frequency of 200 Hz, according to our hardware specifications.

Up to 90 trials were recorded per condition, to guarantee the necessary number of EEG epochs for the analysis. On average, 24 epochs with correct responses were selected for the PSD calculation. In a few cases, it was not possible to select 24 clean epochs (see Table A1 in Appendix A). In those cases, we admitted up to a minimum of 19 clean epochs. This number of epochs assured, on one hand, the smoothness of the PSD and, on the other hand, that the cross-spectral matrix was positively defined (at least as many segments as EEG leads were needed to achieve this condition), a requirement for the successive processing and statistical analyses [67]. Selecting, on average, 24 artifact-free EEG epochs in all subjects, accounted for having balanced samples among all subjects for the processing step, so the results were not likely to be influenced by the epoch choice.

For the PSD calculation, 800 ms of the retention phase from each trial with correct responses was used. The recordings were edited off-line by an expert neurophysiologist, who selected only artifact-free and quasi-stationary epochs around the stimulus onsets, without using any automatic algorithm for artifact rejection. Automatic artifact rejection is useful for high-density EEG recordings, where the visual inspection of the data becomes difficult and controversial, as well as for obtaining a clean recording of all channels over long periods of time. However, since our recordings contained the standard setting of 19 electrodes, we preferred to keep control of the recording conditions to obtain clean recordings and avoid the use of automatic procedures, which are not 100% guaranteed to produce a clean signal and which may also introduce undesirable effects in the “cleaned” signal.

To attenuate the well-known leakage (mixing) problem of the EEG signals at the scalp due to volume conduction [68], we performed our PSD analysis at the estimated primary current sources. For this, we first applied the *s-Loreta* technique [59], which transferred our data from 19 leads to a high-resolution volumetric grid of 3244 sources. However, as stated in Biscay et al. [68], besides the difficulty of analyzing such a high number of sources, there is the limitation that only a small number of sources can be independently estimated for a given number of EEG sensors; specifically, the maximum number of independent sources after solving the inverse problem by any linear method is the number of EEG sensors minus 1. In their paper, they also presented an algorithm that, under quite mild assumptions, can completely unmix the signals for that small number of sources when their domains are specified as corresponding to given regions of interest (ROIs) of said high-resolution grid. In the present paper, we adhered to that methodology.

Before estimating the PSD at the sources by means of *s-Loreta*, the EEG data recorded with the linked earlobes reference were re-referenced to the Average Reference montage. This step addressed two drawbacks: (a) the possible effect induced by possible unequal impedances between the two earlobes, and (b) the primary current estimated at the sources by means of *s-Loreta* was reference free [69].

In the qEEG literature, in order to choose the specific sources (or ROIs) for PSD analysis, it is frequent to use one of the following methods: (1) a selection based on prior alleged knowledge of brain functioning, such as working memory networks previously identified through neuroimaging [41]; (2) a selection of the sources closer to the 10–20 leads, which is not technically arbitrary since the source localization methods are usually more precise in the regions closer to the sensors; (3) a data-driven approach where the ROIs are selected based on the intrinsic variability of the data. The first two methods are not optimal since they ignore the data themselves and do not provide the real brain areas involved in a specific experimental task. In this work, we used a data-driven approach based on the eigenvector centrality mapping technique (ECM) [70] adapted to the present work by the authors.

The ECM is a technique based on the calculation of the principal-components decomposition of a similarity matrix, usually based on the signal in the time domain over all the voxels. It computes its first principal component and interprets each entry of this vector as an index of global connectivity for the corresponding brain voxel. The voxels with higher connectivity indexes are considered the most connected voxels in the brain. In general, the ECM method is calculated for each subject separately, and for group analysis, a statistical test is performed among the subjects to select those voxels with a high connectivity index in most subjects. In our case, we constructed the similarity matrix as the one formed by the absolute values of the correlations between the sources of all voxels. We developed a version of the power method algorithm optimized in terms of memory usage and CPU (Central Processing Unit) intensity, which could obtain the first principal component for all the subjects at the same time. In this way, it was not required to perform a statistical analysis to select the most connected voxels, since the global connectivity index that comes out of our approach is a group index of connectivity, and the voxels with a high global connectivity index will be common for most of the subjects. With this index of global connectivity obtained by the above-described procedure, 18 ROIs (the number of scalp sensors minus 1) were selected. More specifically, not only were the sources identified by this index obtained, but we also included the equivalent sources in the contralateral hemisphere in the cases when they were not selected according to their values of the connectivity index. The reason for this addition was to be able to compare how the homologous sources in both hemispheres participated in the task. Figure 3 shows in red the 18 ROIs selected by the data-driven approach. For a better insight into its configuration, the cortex regions nearest to the positions of the sensors at the scalp are also illustrated in blue. Note that many of the relevant areas detected by our algorithm were far from the sources immediately below the scalp sensors. The source signals at the 18 ROIs were processed by the unmixing algorithm elaborated by Biscay et al. [68]. Then, the segments of unmixed signals of those 18 sources (with 160 time points each) in each condition of all the subjects were transformed to the frequency domain with Fast Fourier Transform. This procedure yielded a source spectrum of 40 frequencies, from 1.25 to 50 Hz (with frequency bins every 1.25 Hz), for every ROI for each subject under each task condition and group.

We performed the statistical analysis of the PSD using this narrow band model of 1.25 Hz frequency resolution up to 50 Hz. Nevertheless, in the Results and the Discussion sections, to consolidate the information and make it easier to understand, we summarize the findings using the classic frequency bands: delta (δ) = 1–4 Hz, theta (θ) = 4–8 Hz, alpha (α) = 8–12 Hz, beta (β) = 12–30 Hz, and gamma (γ) = 30–50 Hz (the upper extreme of the interval is never reached to avoid overlapping). In the case of the gamma band, it is usually reported up to 100 Hz; however, due to our hardware limitations, we report here changes up to 50 Hz, which is considered a lower-gamma band.

Finally, for the main group and task condition comparisons, we used a Linear Mixed Effect Model (LME) [71], in which we tested the possible influence of the IQ and the percentage of correct responses (PCR) as predictor variables, i.e., the group (Ctrl or LD), task load (LL or HL), IQ and two levels of the percentage of correct responses (Bad or Good) as factors at each frequency. Since this model included repeated measures from the two tasks on the same subjects, we included the Subjects as a random factor in the analysis.

To safeguard the statistical significance of our results given the high number of comparisons, the alpha level was corrected using the permutations technique [72]. In all the figures that we show in Section 3.2 with the results for the statistical significance, the two horizontal lines indicate the upper and lower significance thresholds at *p* = 0.05, corrected by permutations.

## 3. Results

According to the comparison of the main characteristics of both groups (see Table 1), the Ctrl and LD groups did not differ in age or gender. For the intelligence measurement, using the Wechsler Intelligence Scale for Children (WISC), the Ctrl children showed a higher full-scale IQ (t = 5.46, *p* < 0.001) and a higher WM index (t = 4.25, *p* < 0.001), compared to the children with LD.

### 3.1. Behavioral Results

To assess the possible differences in performance at the behavioral level in terms of the percentage of correct responses and the reaction time between both the two groups and the two task conditions, we used a Linear Mixed Effect Model (LME) [71], including the IQ as a predictor variable [57]. To account for the repeated measures coming from the two conditions on the same subjects, we included the subjects as a random factor in the analysis. Both groups showed performance differences between the WM task conditions (low-load vs. high-load), with a lower percentage of correct responses (Ctrl: low-load = 93.5 ± 6.2, high-load = 80.7 ± 14.1, *p* = 0.05; LD: low-load = 88.2 ± 9.9, high-load = 69.8 ± 16.6, *p* < 0.0001) and longer response times (Ctrl: low-load = 862.6 ± 223.4, high-load = 1016.3 ± 234.6, *p* = 0.03; LD: low-load = 811.7 ± 205.8, high-load = 998 ± 203, *p* < 0.001) than the high-load condition (see Table A2 in Appendix A). These results show that the high-load condition was indeed more difficult for both groups (see Figure 4).

Figure 5 shows the between-group comparisons (Ctrl vs. LD); the LD group displayed a significantly lower percentage of correct responses than the Ctrl group in the high-load condition (*p* < 0.05), and there were no differences between the groups for the low-load condition (*p* = 0.14) (Figure 6). No significant response-time differences between the groups were found.

The IQ was not significantly different between the groups, either as a main effect or in combination with any other factor: Ctrl, LD or Ctrl-LD.

### 3.2. Power Spectral Density Results

Figure 6 shows the effects on the power spectrum comparison between the conditions (low-load vs. high-load) for each separate group, obtained from the LME model. The Ctrl group showed an extended bilateral increase in the upper-theta and lower-alpha activities at the frontal regions (orbitofrontal gyrus, medial frontal gyrus and middle frontal gyrus) in the high-load compared to the low-load condition. On the other hand, in the high-load condition, the subjects showed a decreased activity in the upper-alpha band bilaterally at the occipital poles (Figure 6, top panel). There were almost no significant changes in the LD group between the two conditions (Figure 6, bottom panel). There was only a very small increment in the high-load condition, with respect to the low-load condition, in the alpha band bilaterally at the inferior frontal gyrus. Note that, although this change did not reach the significance level in the control group, it was at the border of significance.

Upon comparing the groups (Ctrl vs. LD) for each WM load condition, similar differences were found for the two conditions. For the high-load condition (Figure 7), the LD group showed more delta and theta power bilaterally in the occipital pole, the angular gyrus and the superior parietal lobe. This difference was also observed in theta bilaterally at the inferior frontal gyrus, the left middle lateral temporal gyrus and the left superior frontal gyrus. Additionally, the LD group had less gamma activity than the Ctrl group bilaterally at the superior parietal lobe, and the right middle temporal gyrus.

When we tested the difference between the conditions, no significant differences between the groups were found. The meaning of this result is that the Ctrl and the LD groups showed the same differences in the high-load condition and the low-load condition.

There were differences in the PSD related to the task responses. The comparison between the subjects with a low percentage of correct responses (bad responders) and those with a high percentage of correct responses (good responders) also showed significant differences in the PSD. The good responders showed a global effect of increased beta activity in central and posterior regions bilaterally (superior frontal gyrus, superior parietal gyrus, occipital pole and angular gyrus) and increased upper-alpha activity in the left angular gyrus, right superior parietal gyrus and occipital pole. They also showed a decrement in upper-beta activity at the left frontal regions (lateral frontal-orbital gyrus, middle frontal gyrus, medial frontal gyrus and superior frontal gyrus) (Figure 8, top panel).

While the top panel of Figure 8 shows a global effect, there are many combinations of factors that may be interesting to analyze. In the bottom panel of Figure 8, we show the differences in the PSD between the good responders and bad responders from the LD group, during the high-load condition. The good responders showed higher beta activity bilaterally at the angular gyrus, superior parietal lobe, inferior frontal gyrus, superior frontal gyrus and left middle lateral temporal gyrus. Additionally, the good responders showed higher activity in the upper-theta and lower low-alpha bands bilaterally at the superior frontal gyrus and the right angular gyrus. Bad responders showed higher PSD values than good responders in the gamma band on the right hemisphere at the lateral frontal-orbital gyrus, medial frontal gyrus, middle frontal gyrus and middle lateral temporal gyrus.

Finally, we also found a relationship between the Intelligence Coefficient (IQ) and the PSD during the performance of the working memory task. Figure 9, top panel, shows the IQ’s main effect on the PSD. There is a positive effect on the high-beta and gamma bands bilaterally at the superior parietal lobe and on high-beta bilaterally at the angular gyrus.

The IQ had a negative main effect on the PSD in the alpha and beta bands at the right medial and middle frontal gyrus. Additionally, it had a similar effect bilaterally on the beta bands in the superior parietal lobes and the left medial frontal gyrus. Additionally, the IQ had a negative main effect on high-theta and low-alpha bands at the left occipital pole.

In the bottom panel of Figure 9, we show the results of the IQ’s interaction with the group differences (CTRL-LD). The observed pattern here is the same as the one shown in Figure 7. It should be noted that this pattern was very consistent and appeared in all factor combinations where the group differences (CTRL-LD) were present, including combinations with the factor percentage of correct responses.

## 4. Discussion

We aimed to examine the EEG power spectrum density of children with LD (compared to children with good academic achievement) during the performance of a WM task. The ENI test was employed as the main criterion to assign the subjects into either group (LD subjects obtained percentile scores of 16 or fewer points); therefore, the groups differed in the scores for two or three academic domains among reading, writing and mathematics. No statistical differences were found between the groups in age or gender, but they differed significantly in both full-scale IQ and the WM index of the WISC test. This lower IQ and WM index (compared to Ctrl children) was an expected finding in our children with LD [5], and according to our inclusion criteria, we made sure that our children had at least a low–average IQ in order to exclude intellectual disabilities [2]. This result is also compatible with the fact that children with LD obtained a lower WM index than their control peers in the WISC test.

As for the behavioral results of the task, the main difference observed between the groups was the lower percentage of correct responses in the high-load condition for the LD group. This suggests that the task was capable of distinguishing between LD and children with typical development by revealing expected WM shortcomings in children with LD, who showed greater difficulty in memorizing digits than their peers with good academic achievement.

The PSD analysis was performed not in the sensor space but at 18 source regions. An adaptation of the eigenvector centrality mapping (ECM) was used as a data-driven procedure for the selection of ROIs. It allowed us to estimate a populational global index of connectivity for each voxel, thus providing a more robust algorithm for the ROI selection. We consider this data-driven procedure a worthy approach for ROI selection since it does not assume arbitrary and uninformed criteria such as the selection of sources closer to the electrodes, or a putative prior knowledge of brain structure–function relationships, like a pre-specified WM network that could not be applied to LD children with insufficiently mapped task-related neural substrates or needing a possibly different strategy to solve the task. Instead, our ROIs broadly explained the sample variance as active regions present in all the subjects during the WM performance. An advantage of this approach was that many ROIs were arranged at prefrontal areas, with no ROIs selected in the central cortex, i.e., near the placement of the electrodes Cz, C3 and C4 (see Figure 3). This finding coincides with other task-related EEG work that does not recognize central areas, while mainly frontal regions have indeed been shown to be involved in WM performance [24,25,73].

Regarding the PSD between-conditions comparison (low-load vs. high-load) for each separate group, the Ctrl children showed more statistical differences between the conditions, with both a higher upper-theta and a higher lower-alpha power for the high-load condition (compared to the low-load condition) in six prefrontal ROIs, and less occipital upper-alpha power for the high-load condition as well. The LD children only showed a higher alpha power in two ROIs, bilaterally in the inferior frontal gyri for the high load. In young adults, it has been reported that at greater WM loads, they show more theta power [24,25] and more lower-alpha activity [31]. Since frontal regions are engaged in the maintenance and recovery of WM representations [74], following Jensen and Tesche [25], the increase in activity at these frequencies with memory load could be due to a sustained neuronal activity to actively maintain the memory representations. Moreover, a better performance is implied with a higher upper-alpha power or, conversely, with less ERD of upper-alpha [40,41], which is precisely what we found in the low-load condition for our Ctrl children compared with the high-load. However, the LD group did not show this consistent pattern of between-condition statistical differences, which suggests that the LD children could require a greater overall recruitment of neural resources to properly respond to even the easy condition of the task. Thus, they are prone to overly recruiting a poorly selective EEG theta and alpha power irrespective of the demands, coupled with an inefficient upper-alpha ERD cortical activation, which suggests a less specialized neural activity due to their maturational lag [15,75].

The between-group analysis (Ctrl vs. LD) for the high-load condition revealed that the children with LD (compared to control children) had an over-recruitment of theta power in bilateral frontal regions (inferior and superior frontal gyri), in the left medial temporal gyrus, and, together with more delta activity, also in posterior parietal (superior parietal areas and angular gyri) and occipital regions. Conversely, for the high frequencies, the LD children showed a frame of under-recruitment, with less gamma power at both the superior parietal areas together with less high-beta power at the right medial temporal gyrus. As mentioned above, a higher delta activity is involved in states of concentration that imply a sustained level of arousal [17,18,20]; a greater theta power is required for attentional control, being more pronounced in less apt individuals and in conditions that require more effort [24,27,28]; more beta power is related with both subvocal rehearsal [45] and response preparation [17]; and more gamma activity has a role in memory maintenance and the binding of memory representations [49,50]. We hypothesize that since our Ctrl children (for the between-condition comparisons) showed an appropriately higher theta power only for the high-load condition, a less effortful state of attention was required to solve the task. As for the higher delta and theta power at most of the sources for the LD group (compared to controls), given that the task lasted 14 min and the trials of both conditions appeared at random, this required constant vigilance. This task was harder for the LD children, who had to maintain both a greater delta power throughout the task, as an index of sustained levels of concentration, and a greater theta activity, as an index of a more effortful attentional control required to properly remember the items required by the task. In relation to this, other work also reported a higher theta power in dyslexic children that performed a phonological discrimination task [51], a finding also understood as an inefficient usage of attentional control.

Regarding the between-groups power differences for the higher frequencies, this is now an opposite picture of the under-recruitment of temporal beta and mostly gamma power at parietal sites for the LD group. This finding could be understood in the sense that, to achieve the demands of responding to, maintaining and rehearsing the items of the task, the control group showed a greater high-frequency activity overall during the 14 min of the task, while the LD group failed to significantly increase this activity. Based on this result, we could assume that the LD children did not fully achieve the neural maturity to recruit these high-frequency bands, thus requiring a more inefficient mobilization of the low frequencies of delta and theta. A recent review that explores a variety of disorders that include WM impairments [76] has pointed out possible different frames of unusual neural activity that consist of hyper- or hypoactivation. According to our findings, we acknowledge a frame of children with LD with an EEG hyperactivation of low-frequency bands, coupled with a hypoactivation of high-frequency bands. However, we acknowledge that there is a need for more EEG PSD work to continue to explore this idea. Additionally, since WM involves a network of interconnected neural modules [77], and both the theta and gamma bands have been related with WM in a fronto-parietal network [73,78], we also suggest future studies to explore the EEG connectivity of WM in children with LDs.

Regarding the secondary analyses regarding the relationship between the PSD with the levels of responders (good responders vs. bad responders) and the IQ, the good responders showed a pattern of more beta power recruitment in posterior (parietal and occipital) sites and more posterior (left angular gyrus and right occipital pole) upper-alpha power, together with an under-recruitment of left frontal beta and gamma power. In agreement with the EEG work that supports the neural efficiency hypothesis [41], we found less upper-alpha ERD (or a higher upper-alpha power) for the better responders, but this was surrounded by a more global and interesting pattern of antero-posterior EEG differences in which the good responders recruited less left frontal power and more posterior high-frequency power bilaterally. Likewise, the PSD interaction with IQ displayed a similar antero-posterior pattern but with a recruitment of even higher frequencies of beta and gamma power at posterior sites that was correlated with a higher IQ, together with a negative interaction with IQ and an under-recruitment of right frontal alpha and beta power. These mixed patterns of under- and over-recruitment in an antero-posterior fashion would not support a neural efficiency hypothesis in which less neural activation is coupled with a higher IQ and a better performance, since our findings would rather point to a notion of an efficiency achieved in better performers that involves less frontal power but a higher posterior power. These findings surely require further exploration, but they are in agreement with Capotosto et al. (2009) [43], by stating that rather than finding EEG responses compatible with a neural efficiency hypothesis, the EEG activity could involve variable levels of EEG frontal power according to the demands of the task.

We wrap up our main findings as follows: The LD children showed an over-recruitment of the EEG slow bands of delta and theta, involved in conditions of sustained concentration and more effortful attentional requirements, respectively. They also conversely showed an under-recruitment of left temporal beta and parietal gamma activity, bands involved in processes of response preparation, memory maintenance and binding, respectively. Thus, our findings point to a picture of a child with an LD with an intrinsic neurodevelopmental lag [14,15,16] who is not able to mobilize the appropriate neural processes at the adequate high frequencies for the proper functioning of the WM system, hence requiring the slower frequencies as a compensatory mechanism, which implies a more effortful attentional control as an attempt to overcome the demands of cognitive performance. Lastly, our work overcame some relevant problems of the EEG literature by selecting ROIs at the brain sources with a data-driven ECM technique in accordance with their implication in the WM task, an approach that offers advantages over the traditional EEG analysis at the sensor space.

## Figures and Tables

**Figure 1 brainsci-10-00817-f001:**
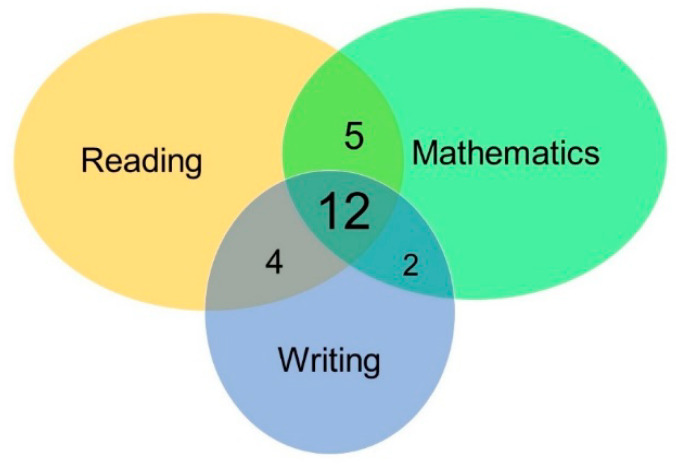
Venn diagram of the frequencies of academic impairments found in our learning disorder (LD) sample: 12 children were impaired in the three domains (reading, writing and mathematics), 5 children were impaired in reading and mathematics, 4 children were impaired in reading and writing, and 2 children were impaired in writing and mathematics.

**Figure 2 brainsci-10-00817-f002:**
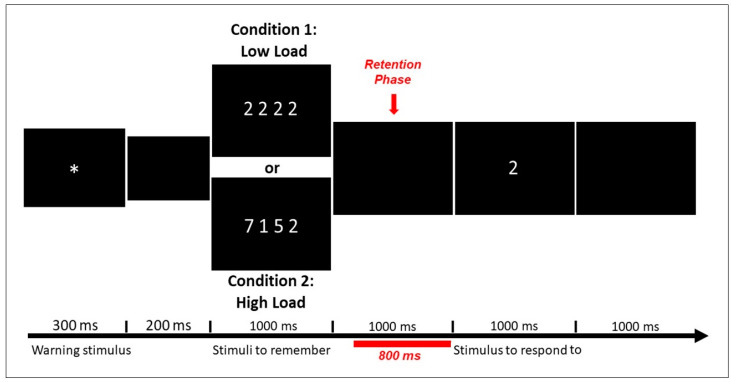
Representation of a single trial (both conditions are represented in the same figure). In this case, the single digit (“stimulus to respond to”) was included previously in the “stimuli to remember” from both conditions, and the subject had to press the button of the “match response”. The segment in red corresponds to the retention phase, the section selected for the power spectrum analysis. The total trial duration was 4500 ms.

**Figure 3 brainsci-10-00817-f003:**
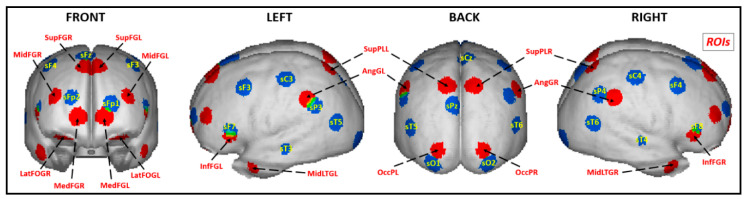
Regions of interest (ROIs) selected by the populational eigenvector centrality mapping (ECM) technique. The sources closer to the 19 leads are in blue, and the 18 ROIs are in red: LatFOGL, Left lateral orbitofrontal gyrus; LatFOGR, Right lateral orbitofrontal gyrus; MedFGL, Left medial frontal area; MedFGR, Right medial frontal area; InfFGL, Left inferior frontal gyrus; InfFGR, Right inferior frontal gyrus; MidFGL, Left medium frontal gyrus; MidFGR, Right medium frontal gyrus; SupFGL, Left superior frontal gyrus; SupFGR, Right superior frontal gyrus; MidLTGL, Left medial temporal gyrus; MidLTGR, Right medial temporal gyrus; SupPLL, Left superior parietal area; SupPLR, Right superior parietal area; AngGL, Left angular gyrus; AngGR, Right angular gyrus; OccPL, Left occipital pole; OccPR, Right occipital pole.

**Figure 4 brainsci-10-00817-f004:**
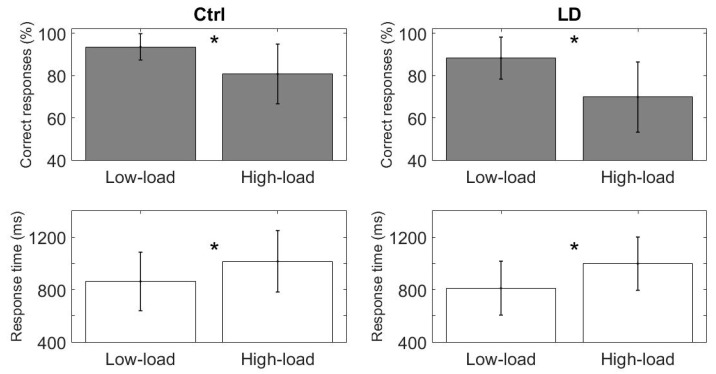
Within-groups behavioral results for the WM task (percentage of correct responses above in gray; response time below in white). The Ctrl group appears on the left; the LD group, on the right. The asterisk indicates statistical differences.

**Figure 5 brainsci-10-00817-f005:**
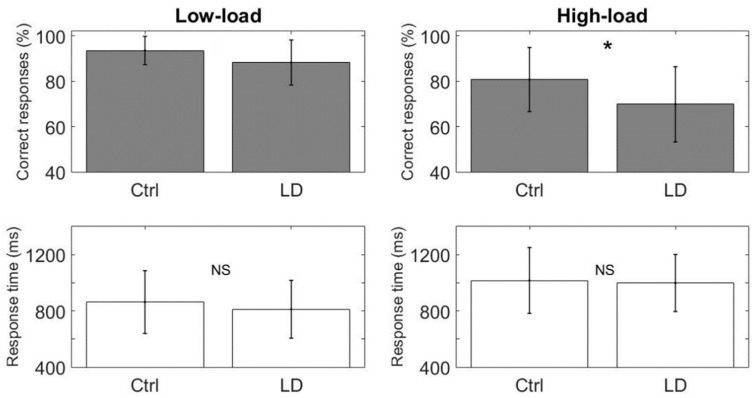
Between-groups behavioral results for the WM task (percentage of correct responses above in gray; response time below in white). The asterisk indicates statistical differences.

**Figure 6 brainsci-10-00817-f006:**
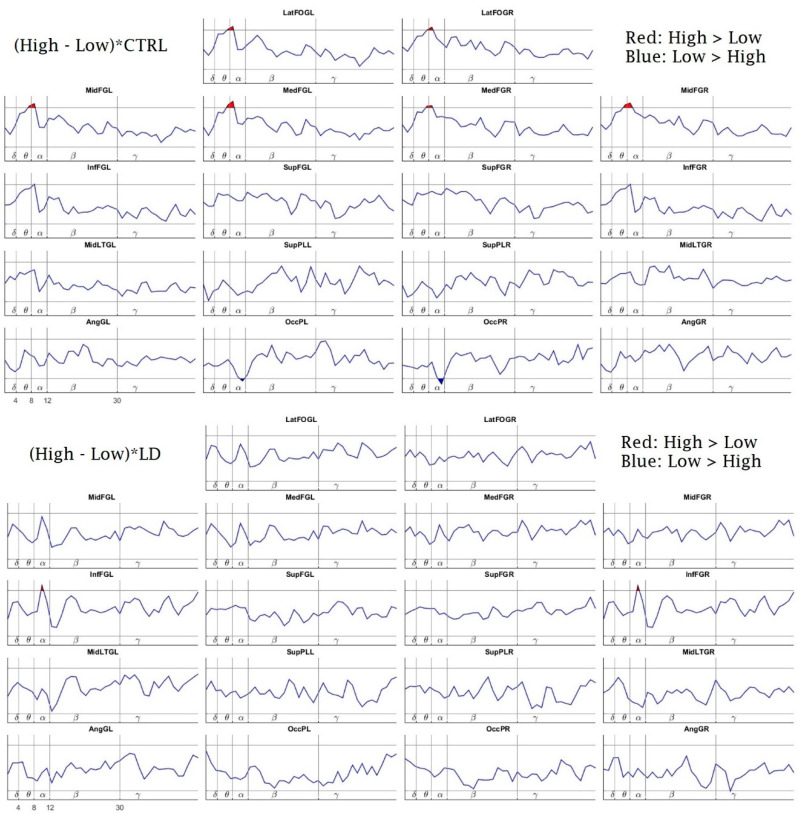
Tasks Effects: Linear Mixed Effects Model Interactions between High Load and Low Load tasks for the CTRL group (top) and LD group (bottom). Power differences between the conditions for each group. The *x*-axis represents the frequencies (1.25–50 Hz), separated by vertical lines into the classic frequency bands: delta (δ) = 1–4 Hz, theta (θ) = 4–8 Hz, alpha (α) = 8–12 Hz, beta (β) = 12–30 Hz, and gamma (γ) = 30–50 Hz (open upper intervals). The *y*-axis represents the t-values of the Linear Mixed Effect Model (LME) procedure. The red patches (above the horizontal lines) indicate a higher power for the high-load condition than for the low-load condition (* *p* < 0.05, randomization-corrected). The blue patches (below the horizontal lines) indicate a higher power for the low-load condition (* *p* < 0.05, randomization-corrected). *LatFOGL/LatFOGR:* Left/right lateral orbitofrontal gyrus; *MedFGL/MedFGR:* Left/right medial frontal area; *InfFGL/InfFGR:* Left/right inferior frontal gyrus; *MidFGL/MidFGR:* Left/right medium frontal gyrus; *SupFGL/SupFGR:* Left/right superior frontal gyrus; *MidLTGL/MidLTGR:* Left/right medial temporal gyrus; *SupPLL/SupPLR:* Left/Rrght superior parietal area; *AngGL/AngGR:* Left angular gyrus; *OccPL/OccPR:* Left/right occipital pole.

**Figure 7 brainsci-10-00817-f007:**
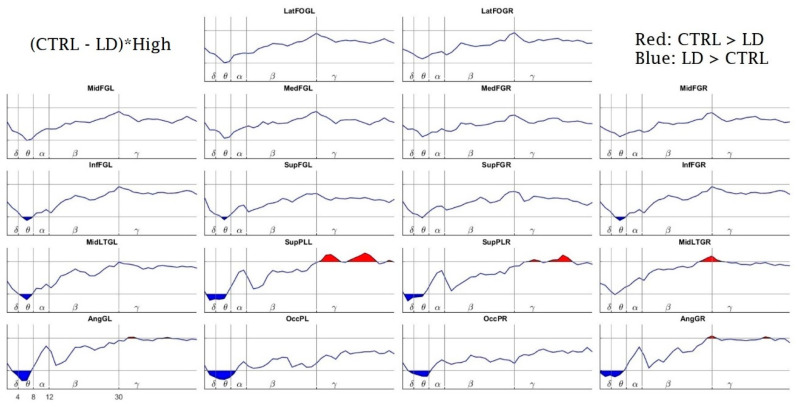
Linear Mixed Effects Model Interactions between CTRL and LD for the High Load task. The *x*-axis, *y*-axis and name of the structures are the same as in Figure 6. The red highlighted areas (above the horizontal lines) indicate a superior power for the Ctrl group in comparison with the LD group (* *p* < 0.05, randomization-corrected). The blue values (below the horizontal lines) indicate a superior power for the LD group (* *p* < 0.05, randomization-corrected).

**Figure 8 brainsci-10-00817-f008:**
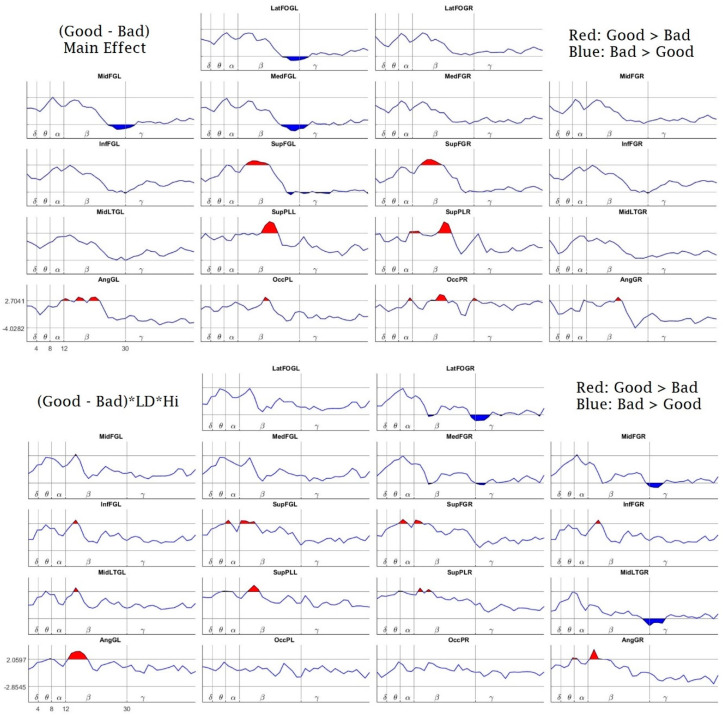
Responders Effects: Linear Mixed Effects Model Main Effect of Responders (top) and the interaction between Good and Bad Responders in the LD group, for the High Load task (bottom). The *x*-axis, *y*-axis and name of the structures are the same as in Figure 6. The red highlighted areas (above the horizontal lines) indicate a superior power for the Ctrl group in comparison with the LD group (* *p* < 0.05, randomization-corrected). The blue values (below the horizontal lines) indicate a superior power for the LD group (* *p* < 0.05, randomization-corrected).

**Figure 9 brainsci-10-00817-f009:**
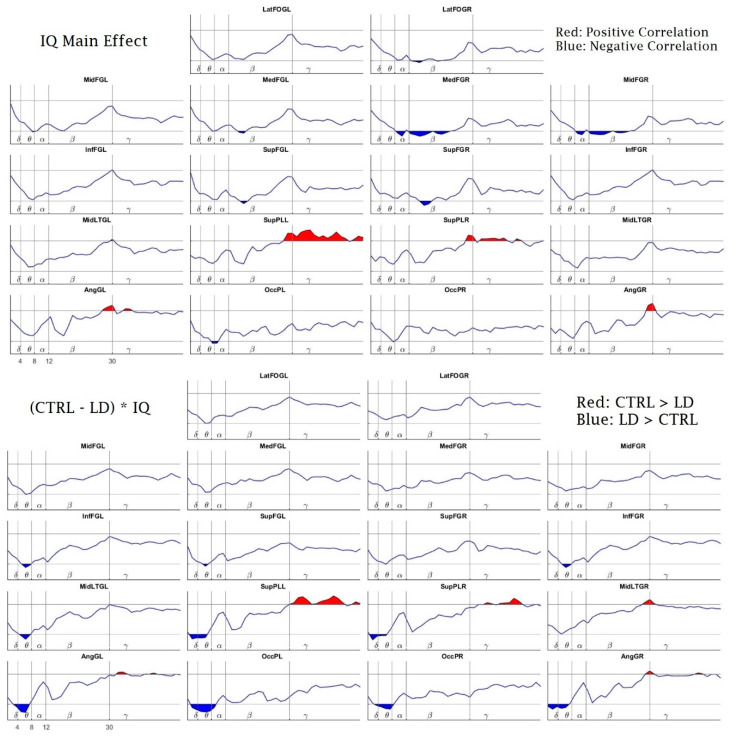
Effect of the Intelligence Quotient (IQ) on the subjects’ EEG spectra. The top panel shows the global effect of IQ, while the bottom panel shows the IQ’s interaction with the difference of CTRL-LD. The “*” sign is the common notation used to identify interactions in the Linear Mixed Effects (LME) models, meaning the interaction between the first and the second factors accompanying the “*” sign. [(CTRL − LD) * IQ]. The *x*-axis, *y*-axis, name of the structures, and red and blue patches are the same as in Figure 6.

**Table 1 brainsci-10-00817-t001:** Sample composition.

	Ctrl Group *n* = 22		LD Group *n* = 23		Statistical Differences between Groups
	Mean	Sd	Mean	Sd	
Age	9.5	0.9	9.4	1.10	t = 0.31 (NS)
WISC test:					
Full Scale IQ	109.3	16.4	88.5	7.9	t = 5.46, *p* < 0.001
Working Memory Index	105.7	16.5	89	8.8	t = 4.25, *p* < 0.001
Female/Male ratio	14/8		12/11		OR = 0.62; CI: (0.18, 2.05); (NS)

Note: (NS): Not Significant; OR: Odds Ratio; CI: Confidence Interval; IQ: Intelligence Quotient; WISC: Wechsler Intelligence Scale for Children.

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
