# Peer review of "Working Memory in Children with Learning Disorders: An EEG Power Spectrum Analysis"

_brainsci, 2020, doi:10.3390/brainsci10110817_

Round 1
Reviewer 1 Report
This study examined neural correlates of cognitive performance in a high vs. low load working memory task in a group of children with learning disorders in comparison with an age-matched control group. Reference-free sLORETA method was implemented and 18 regions of interests (ROIs) were chosen to derive EEG spectral power measures across the EEG frequency bands.
The research report is timely and the findings contribute to the literature on learning disorders. The technical justifications for adopting the analysis and ROI selections are well described. However, the manuscript has several theoretical and technical aspects that can benefit from a major revision to improve its clarity.
- Exactly what EEG frequency bands would show between-group differences was not explained clearly in the Introduction. There also needs to be explicit hypothesis statements for the research questions. The literature review appears thin. For instance, multiple studies have shown cognitive performance impairment associated with not only lower frequency bands but also high frequency activities such as gamma. What are the expected EEG power patterns for the between-group differences in the various frequency bands?
- It was mentioned that "several" of the children in the LD group had attentional issues. How many? Could the inclusion of these children partly account for the differences in percent accurate responses even in the low-load working memory task as shown in Figure 5?
- EEG recording using linked earlobe could have issues with unequal impedance in the two earlobes. The description seems odd here: "The EEG data was sampled every 5 ms and edited off-line with a sampling frequency of 200 Hz." The original sampling rate was 200 Hz; and if so, why is off-line editing done at the same rate? Generally, the sampling rate for original EEG recording would be higher than 200 Hz, and the recording bandwidth would be larger than 0.1-50 Hz.
- One major concern is the use of the criteria "at least 19 artifact-free trials" with correct responses and the lack of description for pre-processing steps. Exactly what was done to get the artifact-free trials? Was independent component analysis implemented to remove artifacts? As the correct response rates were significantly different between the two subject groups, there exists a lack of balance of EEG trials with correct responses, which could influence the results.
- The description of statistical analysis needs to have more details. Why did the researchers report only the t-test results for Figures 4 and 5? Multi-factorial between-group comparison analysis would seem more appropriate.
- Figures 6 and 7 highlight the power differences within group for the high vs. low load tasks and between the two groups for each task. Again, the EEG power data could be extracted for each target frequency band for a full factorial statistical analysis involving repeated measures.
- Although the title of the paper includes the wording about neural correlates, there was no analysis to indicate the existence of significant correlations. There seems to be a gap here by just stating the between-group differences in the behavioral tasks and the between-group differences in the EEG power in certain frequency bands. To account for or control the contributions of IQ and brain-behavior correlates, mixed-effects models can be implemented to include multiple predictor variables. See example here https://www.mdpi.com/2076-3425/7/3/26 Categorical variables such as existence of attention problem in several subjects can be coded in the mixed effects model to see whether there was a significant contribution.
- The EEG PSD results do not necessarily indicate the amount of recruitment of neural resources as the EEG measures reflect the composite excitatory and inhibitory activities. While greater efforts for the high-load memory task are clearly shown in the behavioral response time for both subject groups, there is no direct behavioral evidence to support the claims about attentional/effort differences between the two groups for each task (See Figure 5.)
- Hypotheses based on the neural efficiency notion need to be described more clearly in the introduction. How would neural efficiency be reflected in the different EEG frequency bands across different task demands?
Author Response
The research report is timely and the findings contribute to the literature on learning disorders. The technical justifications for adopting the analysis and ROI selections are well described. However, the manuscript has several theoretical and technical aspects that can benefit from a major revision to improve its clarity.
- Exactly what EEG frequency bands would show between-group differences was not explained clearly in the Introduction. There also needs to be explicit hypothesis statements for the research questions. The literature review appears thin. For instance, multiple studies have shown cognitive performance impairment associated with not only lower frequency bands but also high frequency activities such as gamma. What are the expected EEG power patterns for the between-group differences in the various frequency bands?
Thank you for this enriching point. We have updated the literature review; we added the gamma band to our analyses and clarify the expected EEG patterns. These new changes can be found throughout all the sections. For instance, a modified paragraph now reads:
The previous works generally point to a greater recruitment of delta and theta power in more difficult task conditions, and in less mature (or more unfit) populations with greater difficulties to perform cognitive tasks. Conversely, according to the findings in healthy adults, the EEG indices of a normal WM functioning would imply the recruitment of the higher frequencies of beta and gamma.
- It was mentioned that "several" of the children in the LD group had attentional issues. How many? Could the inclusion of these children partly account for the differences in percent accurate responses even in the low-load working memory task as shown in Figure 5?
Answer: Thank you for this observation. The way this sentence was written in the paper was confusing. Our intention was to highlight that although some of these children failed to correctly answer some items of the checklist of the DSM-V which evaluate the attentional area, none of them was so severe as to be classified as ADHD.
We only applied the DSM-V evaluation to assess whether the children would fall into ADHD or
LD, but we did not apply a finer quantitative scale to measure attention, which would provide
us a range of severity in the scale. Our design was rather focused on the working memory, so we concentrated in designing two tasks which could objectively differentiate a low and high working memory load conditions.
In fact, analyzing the number of correct responses and the reaction time was only an objective way to demonstrate that our two tasks were well designed, particularly that there were significant differences between the two tasks to guarantee that would require different amounts of resource from the subjects to answer them.
It should be emphasized that the further research step of exploring possible factors (such as attentional factors and maybe others) that might mediate and partially account for observed differences in accurate responses and EEG patterns between the LD and control groups while performing the working memory task (which is without any doubt another interesting question) was not within the scope of this work from the beginning. To really deal with this further issue, a more appropriate design would have been necessary that carefully states the attentional and maybe other factors to be considered, their measurement by suitable sensitive scales and a greater sample that sufficiently guarantee to be representative of their variations in the populations. Of course, our findings of differences between groups are a basic empirical requirement for such further explanatory endeavor to be worth of. On this basis, we intend to face this additional research line in future works by means of experimental designs adequate for it. Notwithstanding, as a modest preliminary step in this endeavor, we did take your suggestion of considering the influence of the IQ and the percentage of correct responses in our model and we included them in the analysis, as we will explain in the next questions.
Accordingly, we have corrected in the text. In the Introduction, we changed the text like this:
To our knowledge, the WM (task-related) EEG power spectrum has not been explored in children with LD. This is the main goal of the present work. Specifically, we aimed to compare the EEG PSD of children with LD, with a control group of children with good academic achievement during a Sternberg-type WM task. Additionally, with the purpose of assessing whether the Intelligence Coefficient (IQ) may influence the percentage of correct responses given by the children during the experiment we include the IQ in the analysis as a multifactorial predictor.
We then also characterized the brain electrical activity (PSD) during the task in terms of the differences between the two groups in each condition as well as the differences within each group separately for the two conditions. We performed this analysis using a multifactorial model, in which we included both the IQ and the percentage of correct responses (PCR) as multifactorial predictors [39]. In the case of the PCR, to guarantee the statistical requirement of balanced samples for this analysis, we divided the subjects in two groups according to their PCR as: Good Responders (GR subjects), those with the highest percentage who fall within the top half of the PCR histogram; and Bad Responders (BR subjects) with those ones falling in the lower half of the PCR histogram.
We do not aim in this work to include other possible neuropsychological variables that may influence the children’s response, since it is beyond the said focus of the present work.
And, in Materials and Methods, it reads like this:
Several of them failed on different items in the attentional evaluation of the DSM-V, as it is common in this disorder [45, 46], but none was as severe as to be classified as ADHD according to the DSM-V criteria [2].
- EEG recording using linked earlobe could have issues with unequal impedance in the two earlobes. The description seems odd here: "The EEG data was sampled every 5 ms and edited off-line with a sampling frequency of 200 Hz." The original sampling rate was 200 Hz; and if so, why is off-line editing done at the same rate? Generally, the sampling rate for original EEG recording would be higher than 200 Hz, and the recording bandwidth would be larger than 0.1-50 Hz.
Again, thank you for the observation. We have clarified the writing in the text. Some of our choices are due to hardware restrictions, which did not allow higher sampling frequency. But we did extend our analysis up to 50 Hz to include Gamma band in the analysis, as it will be seen in the answers to the next questions. We have clarified this in the text like this:
Children were comfortably seated in a dim lit faradized and soundproofed room. The EEG was recorded during the task performance using a Medicid IV system (Neuronic Mexicana, S.A.; Mexico) and Track Walker TM v5.0 data system, from 19 leads of the 10–20 system (ElectroCap, International Inc.; Eaton, Ohio) referenced to the linked earlobes (A1–A2). The amplifier bandwidth was set between 0.1 and 50 Hz. The signal was amplified with a gain of 20,000 with electrode impedances at or below 5 kΩ. The EEG data was sampled every 5 ms with a sampling frequency of 200 Hz, according to our hardware specifications.
…
Before estimating the PSD at the sources by means of s-Loreta, the EEG data recorded with the linked earlobes reference is re-referenced to the Average Reference montage. This step solves two drawbacks: a) the possible effect induced by possible unequal impedances between the two earlobes; and b) the primary current estimated at the sources by means of s-Loreta is reference free [51].
- One major concern is the use of the criteria "at least 19 artifact-free trials" with correct responses and the lack of description for pre-processing steps. Exactly what was done to get the artifact-free trials? Was independent component analysis implemented to remove artifacts? As the correct response rates were significantly different between the two subject groups, there exists a lack of balance of EEG trials with correct responses, which could influence the results.
We have clarified the writing in the text. Now it reads like this:
Up to 90 trials were recorded per condition, to guarantee the necessary number of EEG epochs for the analysis. In average, 24 epochs with correct responses were selected for the PSD calculation. In few cases, it was not possible to select 24 clean epochs (see Table I in the Appendix). In those cases, we admitted up to a minimum of 19 clean epochs. This number of epochs assures in one hand, the smoothness of the PSD and, on the other hand, that the cross-spectral matrix is positively defined (at least as many segments as EEG leads are needed to achieve this condition), a requirement for the successive processing and statistical analyses [48].
Selecting in average 24 artifact-free EEG epochs in all subjects, accounts for balanced samples among all subjects for the processing step, so the results are not likely to be influenced by the epochs choice.
Table I. Number of selected epochs by Task and Group.
|
|
LD (# of selected epochs) |
CTRL (# of selected epochs) |
||||||||||
|
19 |
20 |
21 |
22 |
23 |
24 |
19 |
20 |
21 |
22 |
23 |
24 |
|
|
Low Load |
1 |
0 |
0 |
0 |
1 |
17 |
1 |
2 |
0 |
1 |
1 |
16 |
|
High Load |
0 |
0 |
1 |
0 |
1 |
17 |
0 |
0 |
0 |
1 |
0 |
20 |
For the PSD calculation, 800 ms of the retention phase from each trial with correct responses were used. The recordings were edited off-line by an expert neurophysiologist, who selected only artifact free and quasi-stationary epochs around the stimuli onsets, without using any automatic algorithm for artifact rejection. The automatic artifact rejection is useful in high-density EEG recordings, where the visual inspection of the data becomes difficult and controversial, as well as to obtain a clean recording of all channels during long periods of time. However, since our recordings contain the standard setting of 19 electrodes, we prefer to keep control of the recording conditions to obtain clean recordings and avoiding the use of automatic procedures, which are not 100% percent guaranteed to produce a clean signal and which may also introduce undesirable effects in the “cleaned” signal.
- The description of statistical analysis needs to have more details. Why did the researchers report only the t-test results for Figures 4 and 5? Multi-factorial between-group comparison analysis would seem more appropriate.
According to the reviewer suggestion, we made a multifactorial analysis of these variables, including the effect of the IQ on the percentage of correct responses in addition to the group and task load factors. We carry out high resolution analysis in the frequency domain for a fuller description of patterns of EEG differences, which also allows for better detection of differences that are borderline between standard frequency bands. To avoid inflation of the significance level when testing multiple contrasts at different frequencies in such multifactorial model, thresholds by means of permutation techniques were applied. The results are basically like the ones we had first reported, but now the percentage of correct responses between the two groups in the Low-Load task is not significant. Section 3.1 (Behavioral results) has been completely rewritten accordingly.
- Figures 6 and 7 highlight the power differences within group for the high vs. low load tasks and between the two groups for each task. Again, the EEG power data could be extracted for each target frequency band for a full factorial statistical analysis involving repeated measures.
The analysis methodology for the EEG spectra was also changed to a Linear Mixed Effect model, which included the IQ as a predictor variable as well as the percentage of correct responses. The analysis was extended to 50 Hz, to assess changes in the Gamma band. We keep high resolution analysis in the frequency domain for a fuller description of patterns of EEG differences, which also allows for better detection of differences that are borderline between standard frequency bands. To avoid inflation of the significance level when testing multiple contrasts at different frequencies in such multifactorial model, thresholds by means of permutation techniques were applied. There were only few changes in the results shown in Figures 6 and 7 although the IQ did show significant interactions with the EEG. The figures were updated in consequence, and new figures showing the IQ and percentage of correct responses interactions with the EEG were added. The whole section 3.2 was modified accordingly.
- Although the title of the paper includes the wording about neural correlates, there was no analysis to indicate the existence of significant correlations. There seems to be a gap here by just stating the between-group differences in the behavioral tasks and the between-group differences in the EEG power in certain frequency bands. To account for or control the contributions of IQ and brain-behavior correlates, mixed-effects models can be implemented to include multiple predictor variables. See example here https://www.mdpi.com/2076-3425/7/3/26 Categorical variables such as existence of attention problem in several subjects can be coded in the mixed effects model to see whether there was a significant contribution.
We have modified the title of the paper to “Working Memory in Children with Learning Disorders: An EEG Power Spectrum Analysis”, to reflect more accurately our main goal in this work.
We have calculated a linear mixed effects model, with multiple predictor variables, where we have included the IQ and the percentage of correct responses. We have clarified in question 2 the confusion about the attentional problems and why we did not include the analysis of this variable, since it was out the original scope of the paper and our experiment design does not accommodate well to analyze this factor. We at the same time, agree that this can be the subject of a future work.
According to this, we have made changes in the methodology, the results, and the discussion. We have included new figures with the new results, and the discussion reflects all these changes.
We have amended the text on page 3, accordingly to this:
Additionally, with the purpose of assessing whether the Intelligence Coefficient (IQ) may influence the percentage of correct responses given by the children during the experiment we include the IQ in the analysis as a multifactorial predictor.
We then also characterized the brain electrical activity (PSD) during the task in terms of the differences between the two groups in each condition as well as the differences within each group separately for the two conditions. We performed this analysis using a multifactorial model, in which we included both the IQ and the percentage of correct responses (PCR) as multifactorial predictors [39]. In the case of the PCR, to guarantee the statistical requirement of balanced samples for this analysis, we divided the subjects in two groups according to their PCR as: Good Responders (GR subjects), those with the highest percentage who fall within the top half of the PCR histogram; and Bad Responders (BR subjects) with those ones falling in the lower half of the PCR histogram.
Please, check changes on the methodology and the new section 3.2 in Results.
Regarding the suggestion of working by frequency bands, we prefer to continue working with a finer frequency bins, since bands may be less sensible to changes, especially in cases like ours, where the activations occur in the boundaries of two frequency bands, for example, Theta and Alpha. In such cases, half of the frequencies in the band show significant changes and the other half not. Summarizing by bands takes the average of the band, attenuating the significant effect in many cases.
- The EEG PSD results do not necessarily indicate the amount of recruitment of neural resources as the EEG measures reflect the composite excitatory and inhibitory activities. While greater efforts for the high-load memory task are clearly shown in the behavioral response time for both subject groups, there is no direct behavioral evidence to support the claims about attentional/effort differences between the two groups for each task (See Figure 5.)
We agree with this point. There is no direct behavioral evidence to support the attentional/effort statements. However, we do acknowledge that the mentioned patterns of EEG activity work as an index, or biomarker, of such attentional and effort notions; that does correspond with what the EEG PSD is assumed to mean in our cited EEG works. We have modified some wording in our discussion to distinguish the assumptions more accurately from the direct behavioral evidence. For instance, the last paragraph, which contained the original “resources” statement, now reads like this:
We wrap up our main findings as follows: The LD children showed an over-recruitment of the EEG slow bands of delta and theta, involved in conditions of sustained concentration and more effortful attentional requirements, respectively. They also conversely showed an under-recruitment of left temporal beta and parietal gamma activity, bands involved in processes of response preparation, memory maintenance and binding, respectively. Thus, our findings point to a picture of a child with LD with an intrinsic neurodevelopmental lag [14, 15, 80] who is not able to mobilize the appropriate neural processes at the adequate high frequencies for the proper functioning of the WM system; hence requiring the slower frequencies as a compensatory mechanism that imply a more effortful attentional control as an attempt to overcome the demands of cognitive performance. Lastly, our work overcomes some relevant problems of the EEG literature by selecting ROIs at the brain sources with a data-driven ECM technique in accordance with their implication in the WM task, an approach that offers advantages above the traditional EEG analysis at the sensors space.
- Hypotheses based on the neural efficiency notion need to be described more clearly in the introduction. How would neural efficiency be reflected in the different EEG frequency bands across different task demands?
We have taken a step back with the neural efficiency hypothesis according to a more thorough description in the introduction and given our new corrected PSD findings.
The introduction now takes into consideration this aspect:
a pronounced decrease in power of the upper-alpha band, also known as an event-related desynchronization (ERD), has been established to reflect cortical activation instead of inhibition [34, 35]. The upper-alpha ERD has also been related with the neural efficiency hypothesis, which states that a more efficient brain functioning involves less and more focused brain activation since relationships have been observed in healthy adults between a higher IQ and an adequate task performance that involves less neural activity, including a lower brain hemodynamic response [36, 37]. In WM tasks, a better performance has been related with a higher upper-alpha power, or conversely, with less cortical activation indicated by fewer levels of ERD [38, 39]; but there have also been results that doesn´t support this hypothesis, showing the opposite pattern of an upper-alpha ERD [40] and even a higher theta power in brighter individuals [41, 42][41, 42]...
To our knowledge, the WM (task-related) EEG power spectrum has not been explored in children with LD. This is the main goal of the present work. Specifically, we aimed to compare the EEG PSD of children with LD, with a control group of children with good academic achievement during a Sternberg-type WM task. However, children with LD constitute a very heterogeneous population regarding their WM functioning [11]; therefore, we also included the IQ in the analysis as a multifactorial predictor to assess whether the Intelligence Coefficient (IQ) may influence the percentage of correct responses given by the children during the experiment.
We characterized the brain electrical activity (PSD) during the task in terms of the differences between the two groups in each condition and as the differences within each group separately for the two conditions. We performed this analysis using a multifactorial model, in which we included both the IQ and the percentage of correct responses (PCR) as multifactorial predictors [55]. In the case of the PCR, to guarantee the statistical requirement of balanced samples for this analysis, we divided the subjects in two groups according to their PCR as: Good Responders (GR subjects), those with the highest percentage who fall within the top half of the PCR histogram; and Bad Responders (BR subjects) with those ones falling in the lower half of the PCR histogram. Previously mentioned findings (in healthy adults) about the neural efficiency hypothesis have identified a relationship between less upper-alpha ERD with a higher IQ and a better cognitive performance of good responders [37, 56]. Therefore, we also examined the relationship between the PSD, the type of responders and the IQ considering this neural efficiency hypothesis.

Reviewer 2 Report
In this interesting paper authors undertake the problem of working memory at children with learning deficit. They use the source localisation method and observe the power spectrum of particular brain regions of interest. Conclusions are presented in a clear, satisfactory way. The number of electrodes was typical, it would be interesting to observe the results using dense array EEG. The same applies to the quantity of subject and control groups -- less than 30. Maybe in future research?
Author Response
Dear Reviewer
Thank you very much for your very positive opinion.
Yes, a denser electrodes array would benefit the work since it would allow to estimate more unmixed signals at the sources and give information about more brain regions.
Also, it is important to be able to collect information from more subjects. In our case, we were limited by the available resources but it is our purpose to continue working in this direction.
Thanks again
Round 2
Reviewer 1 Report
The authors have addressed my concerns adequately.
One minor point: As the EEG data were recording with the bandpass filter of 0.1-50 Hz, please specify that the gamma activity (30 ~ 50 Hz) as analyzed and reported reflects low gramma band oscillations as gamma is typically reported with a wider range such as 30~100 Hz.
Somehow the pdf file for the manuscript has some problems at the end for the appendix and references. But the supplementary file in Word format looks fine.
Author Response
The authors have addressed my concerns adequately.
Thanks to the reviewer for the very fast feedback.
One minor point: As the EEG data were recording with the bandpass filter of 0.1-50 Hz, please specify that the gamma activity (30 ~ 50 Hz) as analyzed and reported reflects low gramma band oscillations as gamma is typically reported with a wider range such as 30~100 Hz.
We have clarified this aspect in the text. In Line 274 it reads like:
We performed the statistical analysis of the PSD using this narrow band model of 1.25 Hz frequency resolution up to 50 Hz. Nevertheless, in the results and the discussion sections, to consolidate the information and make it easier to understand, we summarized the findings using the classic frequency bands: delta (δ)= 1-4 Hz, theta (θ)= 4-8 Hz, alpha (α)= 8-12 Hz, beta (β)= 12-30 Hz, and gamma (γ)= 30-50 Hz (the upper extreme of the interval is never reached to avoid overlapping). In the case of the gamma band, it is usually reported up to 100 Hz; however, due to our hardware limitations, we report here changes up to 50 Hz, which is considered a lower-gamma band.
Somehow the pdf file for the manuscript has some problems at the end for the appendix and references. But the supplementary file in Word format looks fine.
We have checked the new PDF file, and everything is fine.
